# Gastrointestinal Digestion of a Grape Pomace Extract: Impact on Intestinal Barrier Permeability and Interaction with Gut Microbiome

**DOI:** 10.3390/nu13072467

**Published:** 2021-07-19

**Authors:** Diego Taladrid, Dolores González de Llano, Irene Zorraquín-Peña, Alba Tamargo, Mariana Silva, Natalia Molinero, M. Victoria Moreno-Arribas, Begoña Bartolomé

**Affiliations:** Department of Food Biotechnology and Microbiology, Institute of Food Science Research, CIAL (CSIC-UAM), C/Nicolás Cabrera 9, 28049 Madrid, Spain; d.taladrid@csic.es (D.T.); d.g.dellano@csic.es (D.G.d.L.); irene.zorraquin@csic.es (I.Z.-P.); alba.tamargo@csic.es (A.T.); mariana.silva@csic.es (M.S.); natalia.molinero@csic.es (N.M.); victoria.moreno@csic.es (M.V.M.-A.)

**Keywords:** grape pomace, intestinal barrier, gut permeability, tight junction, microbiota, phenolic metabolites, short-chain fatty acids (SCFA)

## Abstract

Grape pomace (GP) is a winemaking by-product rich in polyphenols and fibre. Supplementation with GP extracts has shown potential benefits against oxidative stress- and inflammation-related pathologies. As a new nutritional target, this paper explores the impact of the ingestion of a grape pomace extract on intestinal barrier functionality. A GP extract was sequentially subjected to gastrointestinal and colonic digestion using the dynamic gastrointestinal simulator (simgi^®^). This generated two simulated fluids: intestinal-digested extract (IDE) and colonic-digested extract (CDE). The effects of these two fluids on paracellular permeability and the expression of tight junction (TJ) proteins (i.e., zonula occludens-1 (ZO-1) and occludin) were assessed in Caco-2-cell monolayers grown in Transwell^®^ inserts. The IDE fluid significantly (*p* < 0.001) reduced the paracellular transport of FITC-dextran with respect to the control, whereas no significant differences (*p* > 0.05) were found for CDE, which could be due, at least partially, to the pro-leaky effect of the colonic digestion medium. Accordant slight increases in the mRNA levels of both ZO-1 and occludin were observed for IDE, but without statistical significance. Additionally, the colonic fermentation of the GP extract promoted the production of short-chain fatty acids (SCFA) and phenolic metabolites and led to changes in the relative abundance of some bacteria that might affect paracellular permeability. Overall, this paper reports first trends about the effects of grape pomace extracts on intestinal permeability that would require further confirmation in future experiments.

## 1. Introduction

Grapes are the fourth-most commonly harvested fruit in the world [1], with 57% of all harvests (44.37 million tons) acquisitioned for winemaking. In this process, by-products represent around 20–30% of the total amount of processed grapes [2]. Moreover, the seasonality of wine production produces massive concentrations of by-products in short periods of time, leading to elevated organic loads in effluents and delays in their biologic degradation. Therefore, the accurate management of grape by-products is necessary to promote sustainability and circular economics in agri-food systems. Grape pomace (GP), the solid residue left after must extraction (white grape varieties) or wine fermentation (red grape varieties), is especially rich in polyphenols and dietary fibre, which has made it a promising bioactive ingredient for food products, cosmetics, and more [3]. Recently, GP has been highlighted as a notable food ingredient due to its preservative capabilities [4] and culinary properties [5], which may even enhance consumer popularity and adherence to low-sodium diets for the prevention of hypertension [6]. Several investigations have been successfully executed to evaluate GP’s properties as a modulator of blood pressure, fasting, plasma glucose, serum lipids, and obesity, among others [7,8,9].

The intestinal epithelium acts as a wall that regulates the entrance of unknown or harmful substances from the lumen to the host, so it is often referred to as the “gut barrier” [10,11]. In ordinary conditions, the passage of substances occurs via the transcellular pathway, but some molecules can cross the intercellular space between enterocytes and reach the bloodstream through the paracellular pathway [12]. The integrity between epithelial cells is ensured by the existence of many tight junctions which involve several protein families, including occludins, claudins, junctional adhesion molecules, and plaque proteins [13]. For instance, occludins are involved in the regulation of the leak pathway via interactions with cytoskeleton through plaque proteins, such as ZO-1 [14,15]. Disturbances of tight junctions (TJ) have been associated with the pathogenesis of many diseases [16,17]. In general, impairment of intestinal barrier integrity is known as “leaky gut”, a condition in which several toxic food compounds and bacterial lipopolysaccharides (LPS) reach the bloodstream, leading to endotoxemia and the onset of several gastrointestinal and metabolic diseases, such as inflammatory bowel disease, diabetes, and obesity [18,19,20,21]. Recently, several studies have examined the transformations of polyphenols along the gastrointestinal tract and their impact on the intestinal barrier (see Bernardi et al. [15] for an updated review). Notably, Nunes et al. [22], using a cell model under pro-inflammatory conditions, found that a wine phenolic extract decreased intestinal paracellular permeability, also modulating the expression of key barrier-forming TJ proteins. However, one of the limitations of these studies is that the extracts were used directly without previous intestinal digestion, which might be far from reflecting the reality of the physiological situation. In addition, other authors evaluated in animal models the effect of a feed rich in polyphenols on gut integrity, reporting an enhanced expression of tight junction proteins [23] and prevention of inflammation-dependent gut permeability [24]. In humans, some studies have been carried out indirectly evaluating the effect of polyphenols on gut health by means of their inhibitory activity on zonulin release [15].

From this background, the aim of this investigation was to evaluate the effects of a grape pomace extract (GPE) previously digested in vitro on the gut barrier. For that, a GPE was selected, and its digestion was carried out via the dynamic gastrointestinal simulator (simgi^®^) in two sequential phases to give rise to two fluids, named intestinal-digested extract (IDE) and colonic-digested extract (CDE). As indicators of gut barrier functionality, a Caco-2 intestinal epithelium model was used to assess the effect of the obtained digestion fluids on paracellular transport and TJ protein expression under homeostatic conditions (Figure 1). In parallel, standards of quercetin—which has previously been shown to decrease paracellular permeability under inflammatory conditions [24]—and its metabolite, 3,4-dihydroxyphenylacetic acid (DOPAC) [25], were assayed for comparison. Additionally, changes in microbial metabolic activity and in microbiota composition were evaluated during colonic digestion of the GPE (Figure 1). This paper presents new findings regarding the enhancement of the gut barrier mediated by GPE and the influence of the gut microbiome on that activity.

## 2. Materials and Methods

### 2.1. Grape Pomace Extract

Grape pomace extract (GPE) of the Tempranillo variety was kindly provided by Bodega Matarromera (Valbuena de Duero, Valladolid, Spain). It was obtained via an extraction phase with a hydro-alcoholic solution, followed by centrifugation and concentration, concluding with spray-drying using maltodextrin. The GPE exerted a moderate total phenolic content (47.96 ± 4.08 mg of gallic acid equivalents/g as determined by the Folin-Ciocalteau’s method), highlighting its contents of phenolic acids (1843.03 µg/g), flavan-3-ols (276.31 µg/g), and flavonols (72.76 µg/g) (Table 1). As indicated by the manufacturer, other components were protein (77 mg/g), complex polysaccharides (810 mg/g), dietary fibre (19 mg/g), fat (l < 10 mg/g), and ash (39 mg/g).

### 2.2. Simulated Digestion of the Grape Pomace Extract

The dynamic SIMulator of the GastroIntestinal tract (simgi^®^^)^) (CIAL, Madrid, Spain), previously used for the digestion of red wine polyphenols [26], was used to simulate the digestion of the GPE. In the first phase, the system operated dynamically, simulating gastrointestinal digestion in the stomach and small intestine. Two simulated juices were initially prepared: gastric juice and pancreatic juice. Gastric juice consisted of pepsin (2000 U/mL) (Sigma-Aldrich, St Louis, MO, USA) dissolved in SGF (simulated gastric fluid) [27]. Pancreatic juice consisted of Difco™ oxgall dehydrated fresh bile (6 g/L) (BD, Franklin Lakes, NJ, USA) and pancreatin (4.24 g/L) (Sigma-Aldrich, St Louis, MO, USA) dissolved in SIF (simulated intestinal fluid) [27]; the solution was filtered via a polyether sulfone 0.45 µm-pore membrane.

Once the whole simgi^®^ system was conditioned at 37 °C, the stomach compartment was filled with 65 mL of SGF, and the small intestinal compartment was filled with 55 mL of SIF [27]. Then, the system was fed with 80 mL of a sodium chloride solution (0.1 g/L) containing the extract (12.5 mg/mL, 1 g in total). In the gastric phase, peristalsis was set to 10 s^−1^, the pH curve was decreased from 5.6 to 1.8 via the addition of 6 mL of HCl 0.5 M, and simulated gastric juice was released by a flux of 3.9 mL/min until reaching a total volume of 15 mL. To simulate physiological gastric emptying, transferred flow from the stomach compartment to the small intestine compartment was automatically programmed via the Elashoff function [28]. Following this, a total of 95 mL of stomach efflux was transferred to the small intestine compartment. Simulated pancreatic juice (40 mL) was then added to the small intestine compartment at a flow rate of 5 mL/min. Then, intestinal digestion was carried out over 2 h at 37 °C, pH 7, and 150 rpm under anaerobic conditions. At the end of this phase, the intestinal digested extract (IDE) was obtained, resulting in a final dilution of 1:4 (*v*/*v*) with respect to the feeding solution.

In a second phase, using the simgi^®^, the IDE was subjected to static fermentation (batch culture fermentation that implements the unique addition of a carbon source at the initial time) with faecal microbiota obtained from a healthy volunteer who had not consumed antibiotics during the 6 months prior to the delivery of the sample. Since the microbiota is a live system subjected to great inter-assay variability, fermentations were carried out in triplicate. Each sterilized fermentation flask was filled with 25 mL of colonic nutrient medium [29], then inoculated with 5 mL of the prepared faecal suspension (1 g of faeces in 10 mL of PBS 0.1 M pH 7 as described in a previous study [30]) and finally supplemented with 30 mL of intestinal digested extract (IDE). Colonic fermentations were carried out to simulate the conditions of the distal region of the human large intestine at a pH of 6.8 and a temperature of 37 °C under an anaerobic atmosphere for 48 h [29]. An identical procedure replacing IDE with 30 mL of colon nutrient medium, also in triplicate, was carried out as a fermentation control. Samples were collected at 0, 24, and 48 h. The samples resulting from 48 h of incubation were named as colonic-digested extracts (CDE1, CDE2 and CDE3). Samples for further metagenomic analysis were centrifuged at 10,000 rpm for 10 min at 4 °C and pellets were frozen and kept at −80 °C. Supernatants were aliquoted and kept at −80 °C for further analysis.

### 2.3. Caco-2 Cells Culture

Human epithelial colorectal adenocarcinoma cells [Caco-2 cells obtained from the American Type Culture Collection (ATCC^®^ HTB-37TM)] regularly grown in 75 cm² flasks at confluence were seeded at 1.5 × 10^5^ cells/mL into polycarbonate Transwell^®^ inserts (12 mm Ø, 0.4 µm pore size) (Costar; Corning Incorporated, Kennebunk, ME, USA). They were cultured in high glucose Dulbecco’s modified Eagle’s medium (DMEM) (Biowest, Nuaillé, France) with penicillin/streptomycin (1%, *v*/*v*) (Sigma-Aldrich, St. Louis, MO, USA) and enriched with heat-inactivated fetal bovine serum (FBS) (Biowest) (10%, *v*/*v*) and non-essential amino acids solution (1%, *v*/*v*) (Lonza, Basel, Switzerland). Cells were grown at 37 °C in a humidified 5% CO_2_ atmosphere and the medium was renewed every 3 days. After renewing the medium, transepithelial electrical resistance (TEER) was measured with an Epithelial Volt/Ohm Meter (World Precision Instruments, Sarasota, FL, USA) to ensure that cells reached confluence and differentiation, which occurred at 21 days.

### 2.4. GPE Digested Fluids and Phenolics Incubations with Caco-2 Cells

After reaching differentiation and confluence, cells were incubated with the GPE digested fluids (IDE or CDE) or pure phenolic compounds in two independent assays. In the first experiment, the apical medium of the Transwell^®^ inserts was replaced with 500 µL of intestinal digested extract (IDE) (1:40 *v*/*v* diluted in DMEM), with the intestinal digestion medium (IDM) (simulated pancreatic juice 1:40 *v*/*v* diluted in DMEM), with a quercetin solution (200 µM, in DMEM including 0.004% of DMSO to ensure solubility), or with DMEM (as a control). After that, the cells were incubated at 37 °C for 4 h of simulation time, during which the dietary compounds remained in the small intestine under in vivo conditions.

Similarly, in the second assay, 500 µL of each colonic digested extract (CDE1, CDE2, and CDE3, 1:40 *v*/*v* diluted in DMEM), 3,4-dyhydroxyphenylacetic acid (DOPAC) solution (200 µM in DMEM), colonic digestion medium (CDM) (colon nutrient medium 1:40 *v*/*v* diluted in DMEM), and DMEM (as control) were added to cell monolayers and incubated for 16 h, which simulated the average time that dietary components remain in the colon. For both experiments, incubations were carried out in triplicate and experiments were repeated on three different days.

### 2.5. Paracellular Permeability Assay

When the incubation time was over, the apical chamber solutions of the Transwell^®^ inserts were replaced with 500 µL of a DPBS (Dulbecco’s phosphate-buffered saline) solution containing 1 mg/mL of fluorescein isothiocyanate (FITC-)-dextran 4 kDa, and the basolateral sides of the plate inserts were filled exclusively with DPBS. Then, the plate was incubated at 37 °C for 30 min. Afterward, 100 µL from the basolateral chamber of each well were taken in triplicate to measure the concentration of FITC-dextran via fluorescence intensity in a BioTek FL600 microplate reader (BioTek, Winooski, VT, USA) with an excitation wavelength of 480 nm and an emission wavelength of 520 nm.

### 2.6. Extraction of mRNA Related to TJ Proteins of Caco-2 Cells and RT-PCR

At the end of the experiment, Caco-2 cell monolayers were scraped and withdrawn with 500 µL of cold PBS and then centrifuged at 1500 rpm for 10 min. Next, 350 µL of the RA-1 buffer (with 1% of β-mercaptoethanol) was added to the pellet prior to mRNA extraction carried out with the Nucleospin^®^ RNA XS kit (Macherey-Nagel, Düren, Germany), following the manufacturer’s instructions. cDNA was obtained using the qPCRBIO cDNA Synthesis Kit (PCR Biosystems, Wayne, NJ, USA) according to the procedure described by the manufacturer.

Finally, qRT-PCR was performed, employing a ViiA™ 7 Real-Time PCR System (Applied Biosystems, Foster City, CA, USA) to quantify the genetic expression of the TJ proteins ZO-1 and occludin using the GADP gene as housekeeping. For that, 1 µL of cDNA was amplified in a 10 µL PCR reaction containing 0.5 µL of each primer (500 nM), 5 µL of PowerUp SYBR Green Master Mix (Applied Biosystems), and 3 µL of RNase/DNase-free water. The PCR began with a first cycle at 95 °C for 3 min followed by 55 cycles, each composed of a denaturation step (95 °C, 10 s), an annealing step (55 °C, 30 s), and an elongation step (72 °C, 30 s). Primers employed in the amplification were as follows: ZO-1 F, 5′-GGTGAAGTGAAGACAATG-3′; ZO-1 R, 5′-GGTAATATGGTGAAGTTAGAG-3′; occludin F: 5′-ATGAGACAGACTACACAACTGG-3′; occludin R: 5′-TTGTATTCATCAGCAGCAGC-3′; GADPH F: 5′-TGCACCACCAACTGCTTAGC-3′; and GADPH R: 5′-GGCATGGACTGTGGTCATGAG-3′. A melting curve was employed to ensure the specificity of the amplification products. The mRNA levels of each protein in the different samples were normalized against GADPH as a housekeeping gene and expressed, as the fold increase respective to the control, using the E^−ΔΔCT^ method.

### 2.7. Analysis of Phenolic Metabolites

Prior to analysis, frozen aliquots of CDE1, CDE2, and CDE3 were defrosted and filtered at a pore size of 0.22 µm. The UPLC-ESI-MS/MS method for the determination of phenolic metabolites was described in a previous study [31]. The multiple reaction monitoring mode (MRM) was followed to collect data. Quantification was carried out employing calibration curves of each corresponding standard compared to the internal standard (4-hydroxybenzoic-2,3,5,6-d4 acid). The MS/MS parameters (cone voltage, collision energy and MRM transition) are also described in previous work [32].

### 2.8. Analysis of Short-Chain Fatty Acids

Prior to analysis, frozen aliquots of CDE1, CDE2, and CDE3 were defrosted and filtered at a pore size of 0.22 µm. The analysis of SCFA was carried out following the SPME-GCMS method, in triplicate, according to the methodology described in previous work [26]. SCFA concentration was calculated from the calibration curves of their corresponding standards compared to the internal standard (2-methylvaleric acid).

### 2.9. Bacterial DNA Extraction and Sequencing of 16S Ribosomal DNA in the Colonic Samples

The pellets from the faecal fermentations were subjected to DNA extraction and 16S rDNA sequencing. Initially, several microbial counts were realized. We observed that there were no differences among CDE1, CDE2, and CDE3 at any times of sample collection (0, 24 and 48 h). For that reason, and with the aim of ensuring an optimal concentration of DNA for sequencing, the three pellets were mixed prior to DNA extraction. Then, DNA was purified using the QIAamp DNA Stool Mini Kit (Qiagen, Hilden, Germany), following the manufacturer’s recommended protocol.

The two-step PCR Illumina^®^ protocol was chosen to prepare the libraries, including PCR Blockers in the first process for minimizing the amplification of mitochondrial and chloroplast DNA [33]. The V3-V4 region of the 16S ribosomal RNA gene was amplified using the forward primer CCTACGGGNBGCASCAG and the reverse primer GACTACNVGGGTATCTAATCC. Sequencing was subsequently carried using an Illumina^®^ MiSeq instrument (Illumina^®^, San Diego, CA, USA) with 2 × 500 paired-end reads. Next, it was analysed via amplification and sequencing of the 16S rRNA v3-v4 gene. Raw files are available in the National Center for Biotechnology (NCBI) repository under the project code PRJNA723163.

The DADA2 algorithm was employed for deionisation, alignment of pairs, and the filtering of chimeras in the raw data [34]. The error correction model implemented in this algorithm allows the differentiation of even a single nucleotide, leading to the formation of Amplicon Sequence Variants (ASVs). A total of 201,387 good quality reads were obtained for bacterial DNA. The taxonomic assignment was performed using the naïve Bayesian classifier implemented in DADA2, with Silva v132 as reference database [35] and a bootstrap cut-off of 80%.

### 2.10. Statistical Analysis

All statistical tests were carried out with R. Each treatment tested in Caco-2 cell monolayers was individually compared to the control incubation employing the *t* test (vegan package) for both paracellular transport and TJ gene expression data. In addition, intestinal digested extract (IDE) and colonic digested extracts (CDE1, CDE2, and CDE3) were compared with their respective media (intestinal digestion medium or IDM, and colonic digestion medium or CDM). The *t* test was also employed to compare differences in phenolic metabolites and SCFA between the mean of the three CDEs and the control fermentation at each time of sampling. Alpha diversity indices were calculated using the phyloseq package and the Heatplus and gplots packages were used to show the relative abundance of bacterial genera greater than 1%.

## 3. Results

### 3.1. Impact of the Grape Pomace Extract in Paracellular Permeability and TJ Gene Expression at Small Intestine Level

Figure 2a depicts the effects of the intestinal digestion medium (IDM), intestinal-digested extract (IDE), and quercetin on paracellular permeability across polarized Caco-2 monolayers, after 4 h incubation, as the means of paracellular transport of FITC-dextran from the apical side to the basolateral side of the Transwell^®^ inserts (% respect to control). The Caco-2 cell monolayers seem to be unaffected by the simulated pancreatic juice, since no statistical differences between IDM and the control were found (Figure 2a). However, incubation with the intestinal digestion extract (IDE) led to statistically significant reductions (*p* < 0.01) in the paracellular transport of FITC-dextran in comparison to incubation with the intestinal digestion medium (IDM), revealing its potential protective role on the intestinal barrier under the conditions of the small intestine. Moreover, similarly to IDE, quercetin also significantly reduced paracellular permeability (*p* < 0.001) when compared to the control (Figure 2a).

Regarding the genetic expression of the TJ proteins, after 4 h of incubation with quercetin, slight increases in the mRNA levels of both ZO-1 and occludin were observed, although no statistically significant differences (*p* > 0.05) with respect to the control or the IDM were found (Figure 2b,c). The IDE fluid also exhibited a small tendency to enhance occludin expression and repress ZO-1 expression, again without statistical significance (Figure 2b,c).

### 3.2. Impact of the Grape Pomace Extract in Cell Permeability and TJ Gene Expression at Colon Level

The results from the incubations of Caco-2 cells CDM (colon digestion medium), faecal-fermented fluids (CDE1, CDE2, and CD3), and 3,4-dihydroxyphenylacetic acid (DOPAC) at a longer period of time (16 h) are shown in Figure 3. The colon digestion medium was slightly harmful to the monolayers as it significantly (*p* < 0.05) increased the paracellular transport of FITC-dextran (Figure 3a). Under these conditions, one of the colonic fermented fluids (CDE2) was capable of restoring the damage exerted by the colonic medium as it decreased paracellular permeability compared with the CDM (Figure 3a). Therefore, with respect to the CDM, the differential components of the CDE fluids seemed to counteract the FITC-dextran permeability mitigating the CDM’s adverse effects and leading to non-significant differences. However, all of the remaining digested extracts (CDE1 and CDE3) were different from both the CDM and control samples. In addition, DOPAC enhanced gut barrier integrity by reducing the paracellular transport of FITC-dextran in a significant way (*p* < 0.001) compared to control.

Similarly to what occurred after incubation with the intestinal digested samples, none of the colonic samples or DOPAC exhibited a significant effect on TJ gene expression Figure 3b,c. However, in this case, there was a clear trend of CDE1 and CDE2 to enhance the expression of both ZO-1 and occludin.

### 3.3. Interaction of the Grape Pomace Extract with Gut Microbiota

The interaction of the grape pomace extract with gut microbiota was studied by assessing gut microbiome composition as well as microbial metabolic activity (i.e., the production of phenolic metabolites and SCFA) during faecal fermentation of the intestinal-digested extract (IDE) (Figure 1).

#### 3.3.1. Evolution of the Gut Microbiome

The faecal microbiota used in the simulation of the colonic digestion of the GPE exhibited good indexes for alpha diversity (Observed, Shannon, and Simpson indexes) (Table 1, time 0 h). As expected, bacterial diversity tends to slightly decrease over time (24 and 48 h), this response being more accentuated for colonic digestion in the presence of the intestinal-digested extract (IDE) (Table 2). However, because of the absence of replicates from the sequencing data, these outcomes could not be extrapolated to larger conclusions.

The predominant families in the faecal samples at the initial stage of the fermentations were *Bacteroidaceae*, *Ruminococcaceae*, *Prevotellaceae*, and *Lachnospiraceae* (Figure 4, time 0 h). During both fermentations, in the absence (Figure 4a) and in the presence of the IDE (Figure 4b), *Bacteroidaceae* showed a small decrease in its relative abundance, accompanied by a drastic reduction in *Prevotellaceae*, *Ruminococcaceae*, and *Lachnospiraceae*. In contrast, the empty niche left by those families led to an increase in *Enterobacteriaceae*, *Desulfovibrionaceae*, *Akkermansiaceae*, *Acidaminococcaceae*, and *Veillonellaceae*, this last case being accentuated in the presence of IDE at 24 h, followed by a considerable reduction at 48 h. Therefore, at the end of the fermentations, the microbiome was dominated by *Bacteroidaceae*, *Enterobacteraceae,* and *Desulfovibrionaceae* in both cases, in the absence and presence of IDE, and also by *Veillonellaceae* in the case of fermentation in the presence of IDE.

The dominant genera at the beginning of the fermentations were *Bacteroides*, *Prevotella*, and *Parabacteroides*, common to both fermentations, and *Subdoligranulum* or *Ruminococcus*, depending on whether in the absence or presence of IDE, respectively (Figure 5). Genera belonging to the families described above followed a similar pattern, leading to a reduction in *Bacteroides* that was faster in the presence of IDE, *Prevotella*, *Subdoligranulum*, and *Ruminococcus*, and an increase in *Veillonella*, *Escherichia*/*Shigella,* and *Akkermansia* in both fermentations. However, the last two genera were more accentuated in the fermentations in the absence and presence of IDE, respectively. Additionally, the microbiota in the fermentations in the presence of IDE was increased in *Megasphaera*, while in its absence, the relative abundance of *Acidaminococcus* increased (Figure 5). These results were obtained under static fermentation conditions which facilitate substrate depletion and might explain, for instance, the increase in *Proteobacteria* and *Akkermansia* observed for all fermentations. In spite of this, the presence of IDW was relevant enough to comparatively modify the faecal microbiome.

#### 3.3.2. Microbial Production of Phenolic Metabolites and SCFA

The main phenolic metabolites detected in the colonic-digested samples of GPE were benzoic acids (protocatechuic, 3-*O*-methylgallic, and 4-hydroxybenzoic acids) and phenylacetic acids (3,4-dihydroxyphenylacetic and 4-hydroxyphenylacetic acids) (Figure 6a). The acids 3-*O*-methylgallic and 4-hydroxyphenylacetic were only detected in the faecal fermentations carried out in the presence of the intestinal-digested extract (IDE): at 24 and 48 h for 3-*O*-methylgallic acid and at 48 h for 4-hydroxyphenylacetic acid (Figure 6a). The rest of the compounds were detected at significantly lower concentrations (*p* < 0.05) in the fermentations carried out in the absence rather than in the presence of the IDE, except for 4-hydroxyphenylacetic acid at 24 h of incubation (Figure 6a).

Among the SCFA measured, acetic and butyric acids showed the highest concentrations, followed by valeric acid, and then propionic and isovaleric acids (Figure 6b). As expected, their concentrations increased during faecal fermentation (Figure 6b). In general, this increase in SCFA occurred mainly in the first 24 h, remaining almost stable until the end of the fermentation (48 h). When comparing fermentations in the presence of IDE with respect to its absence, significantly higher concentrations (*p* < 0.05) were found for the most abundant acids, acetic and butyric, at both 24 and 48 h (Figure 6b). Only valeric acid showed a significantly higher concentration at 24 h for the fermentation in the absence of IDE. No significant differences were found for the rest of the acids at any time (Figure 6b).

## 4. Discussion

Apart from governing the flux of nutrients and other substances, the gut barrier provides protection against pathogens and toxins. The appropriate management of tight junctions (TJ) is essential to maintaining the integrity of the gut barrier. Grape polyphenols are susceptible to eliciting optimal TJ via several potential mechanisms. This study addresses, for the first time, the impact of grape pomace extracts on gut barrier functionality by using a Caco-2 intestinal epithelium model (Figure 1). One of the main inputs of our experimental design was to mimic how GP bioactive molecules (and/or their metabolites) would reach the gut epithelium at comparable physiological concentrations, as well as how they would be maintained for an adequate period of time (Figure 1). In this sense, the simgi^®^ simulator was successfully applied to obtain two distinct fluids, IDE and CDE, corresponding to the intestinal and colonic digestions of GPE, respectively. In relation to the cell model, the accuracy of our experimental protocol for the assessment of cell permeability was confirmed by the positive significant results obtained from pure phenolic compounds like quercetin (200 µM). Many studies have demonstrated that quercetin enhances the integrity of cellular models simulating the gut barrier, both in the absence [36,37] or the presence of stressful conditions [24,38,39].

Our results indicated that the intestinal-digested extract (IDE) of grape pomace successfully limited the paracellular flux of FITC-dextran, suggesting some kind of activity over the TJ (Figure 2). Previous work reported similar outcomes in Caco-2 cells incubated with polyphenol-rich extract from propolis [40]. Moreover, a red wine extract [22] and an anthocyanin-rich extract [41] were also found to inhibit permeabilization in HT-29 and Caco-2 cells subjected to inflammatory cytokines, respectively. In addition, other studies have reported protective effects on intestinal permeability provided by quercetin and grape seed procyanidins in animal models, confirming the results extrapolated from in vitro to in vivo situations [23,24].

One would expect that the restriction of paracellular transport found for the IDE after 4 h of incubation with Caco-2 cells should be associated with an increase in TJ expression, but this was not the case (Figure 2). Some studies reporting effective cell permeability reduction by quercetin or phenolic-rich extracts also found unaltered expression levels of ZO-1, occludin, and other TJ proteins [36,39]. Therefore, TJ boosts, mediated by polyphenols without modifying their amounts, might occur by other mechanisms, such as post-translational modifications of many paracellular transport proteins, particularly TJ proteins, which modulate the functional capacities [42] required in TJ assembly and their cytoskeletal associations [43,44,45]. It has shown that grape polyphenols, including quercetin and procyanidins, act on cytosolic kinases, promoting the cytoskeletal associations of TJ proteins [43,44,45]. Similarly, Yang and collaborators [46] demonstrated that grape seed extract reduced permeability via the recovery of AMPK under inflammatory conditions. Many studies suggest that polyphenols act on the NF-κβ signalling pathway, one of the most important mediators of inflammatory status [43,47,48].

Evaluation of paracellular permeability in the colon involved a longer incubation time (16 h). Under these conditions, the microbial-derived metabolite tested (3,4-dihydroxyphenylacetic acid, DOPAC) notably limited the paracellular flux of FITC-dextran. This is a novel finding since, as far as we know, no previous data about this phenolic metabolite has been reported in the literature. In contrast, the colonic-digested extracts (CDE) showed no significant effect on the paracellular transport of FITC-dextran, also exhibiting quite variable results. However, these outcomes were a consequence of the colonic digestion medium (CDM) as it significantly raised paracellular permeability, and CDE2 was capable of restoring its impairment of the intestinal barrier in a significant way (Figure 3). At the colonic level, GP compounds and/or its metabolites might specifically affect the paracellular permeability by modulating the gut microbiota (Figure 1). Thus, bacteria commonly release extracellular vesicles into the gut lumen; these vesicles are composed of proteins, nucleic acids, or lipopolysaccharide (LPS) that interact(s) with immune and epithelial cells, promoting beneficial or detrimental effects on barrier integrity [49,50,51]. On the favourable side is *Akkermansia muciniphila*, whose vesicles have demonstrated an ability to enhance TJ via the activation of AMPk [52,53]. Bacteria whose LPS are involved in endotoxemia are on the other side. LPS is the main component of the outer membrane of Gram-negative bacteria and induces inflammation through the stimulation of Toll-like-receptor 4 (TLR4), which, in turn, generates intracellular responses that culminate in the release of NF-κβ [54]. Some of the bacteria related to irritable bowel disease, which are LPS productors, are *Escherichia*, *Shigella*, *Desulfovibrio*, and some genera belonging to *Veillonellaceae* [55,56,57]. In relation to this, another main input of the experimental design of this study was the evaluation of the gut microbiome composition, as well as the microbial metabolic activity (i.e., production of phenolic metabolites and SCFA) during faecal fermentation (Figure 1). The results revealed an increase in the relative abundance of LPS-producer bacteria such as *Veillonellaceae*, *Escherichia*/*Shigella*, and *Desulfovibrio* (Figure 4 and Figure 5). Therefore, it was likely that Caco-2 cells were subjected to high concentrations of LPS during the 16 h-incubation with the CDEs. This fact could mitigate the potential role of the microbial-derived polyphenols produced during faecal fermentation of GPE (Figure 6). However, gut integrity is also directed by other microbial-related factors like SCFA. The results of SCFA analysis indicated that their production, especially butyrate (Figure 6), was stimulated in the presence of IDE, which was probably associated with the higher relative abundance of *Bacteroides*, *Ruminococcaceae*, and *Akkermansia*, which are considered to be butyrate-producers. It has long been known that butyrate and propionate, two of the SCFA produced in significantly greater quantities in the presence of IDE (Figure 6), promote gut integrity in Caco-2 cell monolayers [58]. Butyrate exerts its beneficial function on the gut barrier by enhancing the genic expression of TJ proteins via the activation of AMPK and inhibition of inflammatory mediators like NF-κβ and TNF-α [59,60]. This response is not exclusive to in vitro experiments; it has been also reported that pigs whose diet was supplemented with butyrate experienced an up-regulation of occludin and ZO-1 [61]. Additionally, SCFA may decrease LPS translocation into the cell [62], which could contribute to minimizing the effect of pathogenic bacteria. All these results, relative to microbiome composition and microbial metabolic activity, could explain the non-significant effect on paracellular permeability found for CDEs, despite the pro-leaky effect observed for the colonic digestion medium (CDM) (Figure 3).

## 5. Conclusions

Taken together, the outcomes of this study show the positive effects of GP on preserving intestinal epithelium integrity. The biological relevance of our data is that a dietary GPE was tested during simulated gastrointestinal digestion to approach the real physiological conditions at the intestinal lumen. In particular, at the level of the small intestine, the extract seemed to reduce paracellular permeability compared to the sample containing only the intestinal-digestion medium (IDM) and slightly increased mRNA levels of both ZO-1 and occludin. At the colonic level, the harmful effect mediated by the colonic digestion medium (CDM) was restored by one of the colonic-digested extracts (CDE2). Additionally, colonic fermentation of the GP extract promoted the production of short-chain fatty acids (SCFA) and phenolic metabolites. However, further experiments would be required to confirm the initial trends of the effects of grape pomace on intestinal permeability observed in this study. On the other hand, the experimental design carried out in this paper, as well as the new data obtained for reference standards (i.e., quercetin and 3,4-dihydroxyphenylacetic acid), may be useful for the study of the nutritional properties of other food ingredients and agri-food products.

## Figures and Tables

**Figure 1 nutrients-13-02467-f001:**
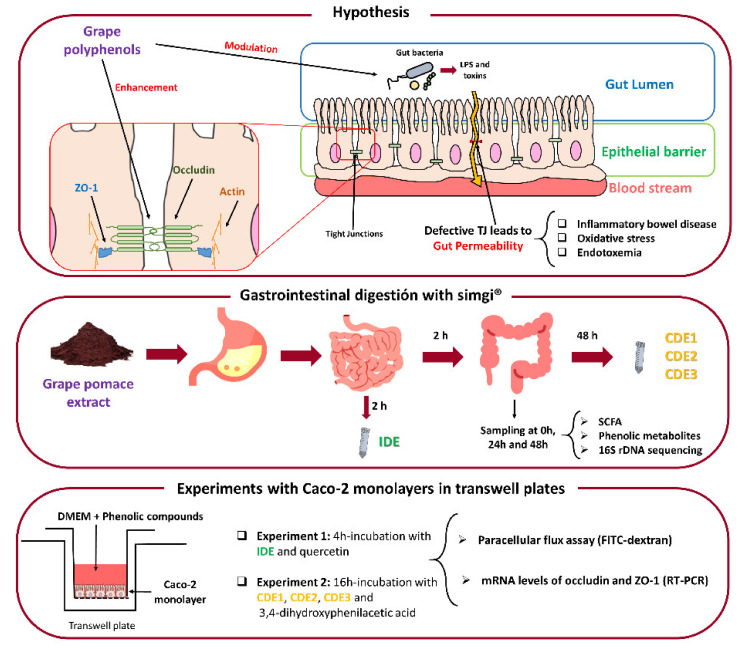
The main objectives of the present study and the experimental design. TJ: Tight junctions. IDE: Intestinal-digested extract. CDE: Colonic-digested extract. SCFA: Short-chain fatty acids. DMEM: Glucose Dulbecco’s modified Eagle’s medium. FITC: Fluorescein isothiocyanate.

**Figure 2 nutrients-13-02467-f002:**
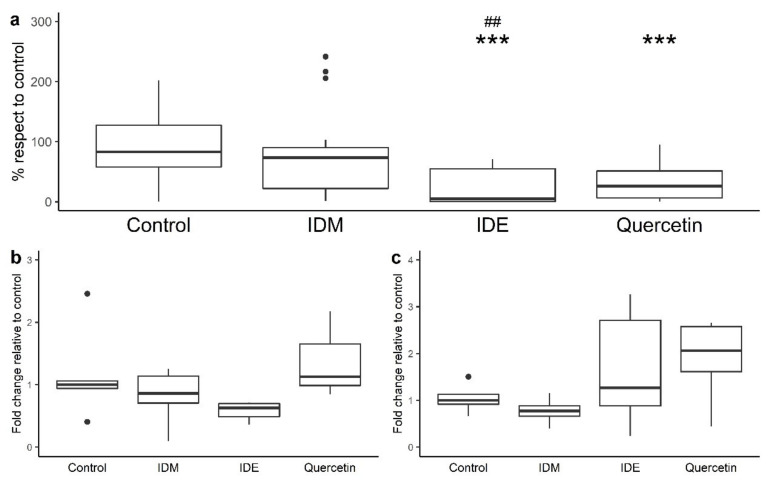
The effects of intestinal digestion medium (IDM), quercetin (200 µM), and intestinal digested extract (IDE) after 4 h incubation on paracellular permeability in Caco-2 cells (**a**) and the genetic expression of ZO-1 (**b**) and occludin (**c**). The *t* test was employed to check the differentiation of each treatment with respect to the control (*** significant differences at *p* < 0.001). Also, the *t* test was employed to check for differences in IDE with respect to IDM (## significant differences at *p* < 0.01). IDM: Intestinal digestion medium. IDE: Intestinal-digested extract.

**Figure 3 nutrients-13-02467-f003:**
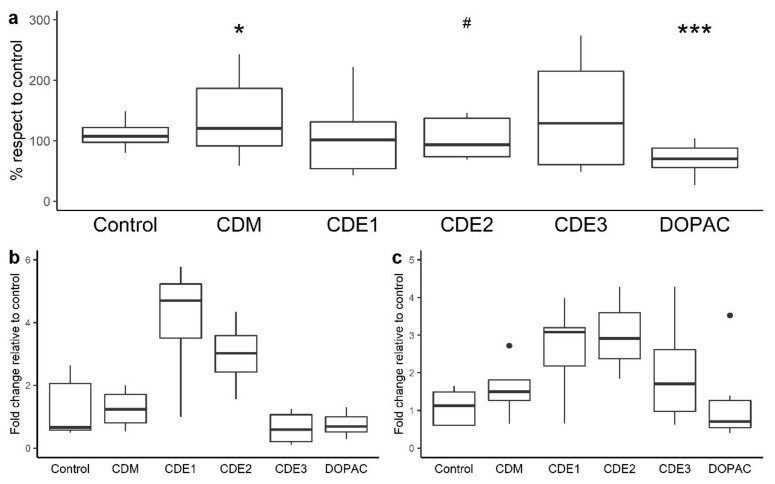
The effects of colonic intestinal medium (CDM), 3,4-dihydroxyphenylacetic acid (200 µM), and colonic digested extracts (CDE1, CDE2, CDE3) after 16 h incubation on the paracellular permeability in Caco-2 cells (**a**) and on ZO-1 (**b**) and occludin gene expression (**c**)**.** The *t* test was employed to check for differences in each treatment with respect to the control (* significant differences at *p* < 0.05; *** significant differences at *p* < 0.001). Also, the *t* test was employed to check for differences in CDEs with respect to CDM (# significant differences at *p* < 0.05). CDM: Colonic digestion medium. CDE: Colonic-digested extract.

**Figure 4 nutrients-13-02467-f004:**
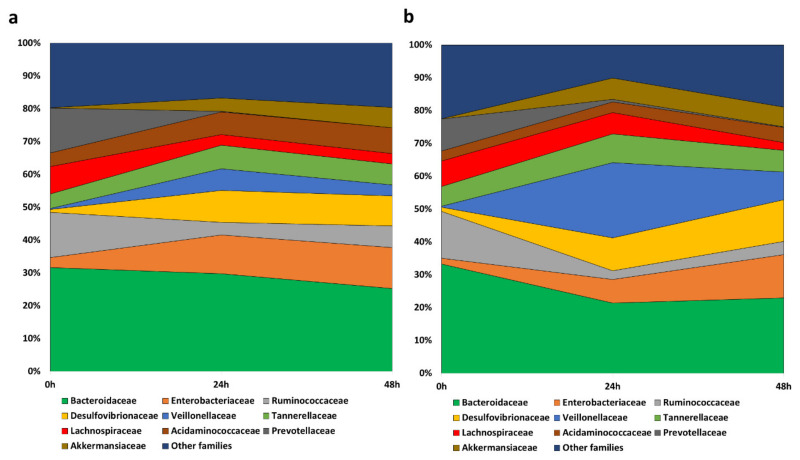
The evolution of bacterial families with relative abundance higher than 5% during faecal fermentations in the absence (**a**) and presence (**b**) of the intestinal-digested extract (IDE).

**Figure 5 nutrients-13-02467-f005:**
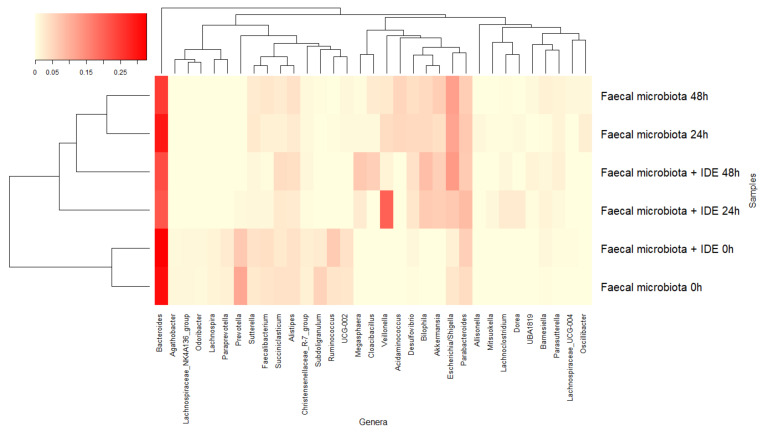
The evolution of bacterial genera with relative abundance higher than 1% during faecal fermentations in the absence and presence of intestinal-digested extract (IDE).

**Figure 6 nutrients-13-02467-f006:**
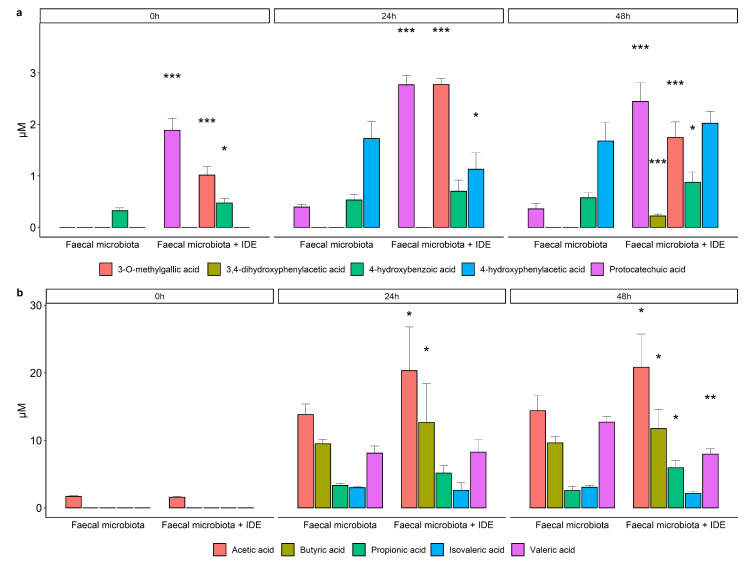
The evolution of phenolic metabolites (**a**) and SCFA (**b**) during faecal fermentations in the absence and presence of the intestinal-digested extract (IDE). Data are means ± SD of the three replicates. Asterisks indicate significant differences with respect to the absence of IDE at the same incubation time: * at *p* < 0.05, ** at *p* < 0.02, *** at *p* < 0.05.

**Table 1 nutrients-13-02467-t001:** The main phenolic compounds in the grape pomace extract (µg/g).

Gallic Acid	Ethyl Gallate	Protocatechuic Acid	(+)-Catechin	(-)-Epicatechin	Kaempferol	Quercetin-3-*O*-glucoside
1772.33	33.04	37.66	205.22	71.08	12.32	60.43

**Table 2 nutrients-13-02467-t002:** The evolution of alpha diversity during faecal fermentation in the absence or presence of the intestinal-digested extract (IDE).

		0 h	24 h	48 h
Observed	faecal microbiota	221	227	194
	faecal microbiota + IDE	241	179	180
Shannon	faecal microbiota	4.25	3.98	3.95
	faecal microbiota + IDE	4.30	3.91	3.76
Simpson	faecal microbiota	0.97	0.96	0.96
	faecal microbiota + IDE	0.97	0.96	0.95

## Data Availability

The data presented in this study are available on request from the corresponding author.

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
