# Peer review of "Gastrointestinal Digestion of a Grape Pomace Extract: Impact on Intestinal Barrier Permeability and Interaction with Gut Microbiome"

_nutrients, 2021, doi:10.3390/nu13072467_

Round 1

Reviewer 1 Report

The authors have addressed comments

Reviewer 2 Report

I have no additional comments.

This manuscript is a resubmission of an earlier submission. The following is a list of the peer review reports and author responses from that submission.

Round 1

Reviewer 1 Report

The authors have included needed language in the material and methods and results section. I still think the data presented  does not support the claims made in the conclusions. It is fine that data is difficult to extrapolate from. I would like the authors to tone down the concluding claims by making it clear that they mostly report interesting trends. A concluding sentence (similarly balanced) should be included in the abstract.

Reviewer 2 Report

Review of nutrients-1262295

Comments to the Authors

Experiments using the simgi® are interesting. However, I could not agree the author’s interpretation of results and conclusion. Especially in the point that I already point out in first round report. Conclusion should be drowned from difference between IDM and IDE or CDM and CDE. Differences between control and IDE or control and CDE could not reflect effect of grape pomace extract. Because IDM or CDM is different from control. Authors showed that control and IDM or control and CDM are not different in several points, but these are not mean control and IDM or control and CDM are same. For example, in Abstract, authors mentioned that “ colonic fermentation of the GP extract promoted production of short-chain fatty acids (SCFA) and phenolic metabolites”. But authors did not compare IDM and IDE. I am concerned about the interpretation of this result.
I understand the authors’ comments about “Since microbiota is a live system subjected to great inter-assay variability, fermentations were carried in triplicate.” (lines 148-151).

Author Response

We thank the reviewer for the comment. In the firstly revised version, we have tried to point out differences between IDM and IDE, but they might have not resulted clearly enough. Taking in mind the new reviewer´s comment, we have done the following changes (highlighted in yellow):

Lines 279-286

The Caco-2 cell monolayers seem to result unaffected by the simulated pancreatic juice since no statistical differences between IDM and the control were found (Figure 2a). However, incubation with the intestinal digestion extract (IDE) led to statistically significant reductions (p<0.01) of the paracellular transport of FITC-dextran in comparison to the incubation with the intestinal digestion medium (IDM), disclosing a potential protective role on the intestinal barrier under small intestine conditions. Moreover, quercetin also reduced paracellular permeability in a significant way (p<0.001) in a similar extent than IDE when both were compared to the control (Figure 2a).

Lines 304-316

Results from the incubations of Caco-2 cells CDM (colon digestion medium), the fae-cal-fermented fluids (CDE1, CDE2 and CD3) and 3,4-dihydroxyphenylacetic acid (DO-PAC) at longer period of time (16 h) is shown in Figure 3. The colon digestion medium was slightly harmful for the monolayers since it significantly (p<0.05) increased the para-cellular transport of FITC-dextran (Figure 3a). Under these conditions, one of the colonic fermented fluids (CDE2) was capable to restore the damage exerted by the colonic medium since it decreased the paracellular permeability compared with the CDM (Figure 3a). Therefore, the differential components of the CDE fluids respect to the CDM, seemed to counteract the FITC-dextran permeability mitigating the CDM adverse effects and leading to non-significant differences. However, any of the remaining digested extracts (CDE1 and CDE3) were different from both the CDM or the control samples. In addition, DOPAC enhanced gut barrier integrity by reducing paracellular transport of FITC-dextran in a significant way (p<0.001) compared to control.

Reviewer 3 Report

Changes have improved the manuscript. Now has been inserted what was requested regarding the implications on patients 

Author Response

We thank the reviewer for his/her revision work.